# Task Challenge and Employee Performance: A Moderated Mediation Model of Resilience and Digitalization

**DOI:** 10.3390/bs13020119

**Published:** 2023-01-31

**Authors:** Irfan Saleem, Tahir Masood Qureshi, Amitabh Verma

**Affiliations:** 1Faculty of Business, Sohar University, Sohar P.O. Box 44 P.C. 311, Oman; 2School of Business & Quality Management, Hamdan Bin Mohammed Smart University, Dubai P.O. Box 71400, United Arab Emirates

**Keywords:** employee resilience, task challenge, employee performance

## Abstract

Lately, organizations are giving attention to enhancing employee resilience due to turbulent economic times caused by lockdowns in the last couple of years. This explanatory research proposes and tests the mediating role of employee resilience to link task challenge and employee performance during the COVID-19 pandemic in government schools of Oman using the broaden-and-build theory. An explanatory research design was used for this empirical study. An Arabic-translated questionnaire version was designed to collect primary data online during the lockdown using simple random sampling. We used the Preacher and Hayes macro to analyze moderated mediation using cross-sectional data and analyzed 441 responses. The findings confirm the mediation roles of employee resilience and moderating roles of digitalization during unusual circumstances. The study has implications for the school administrators, Omani policymakers and schooling staff of the Middle Eastern educational industry.

## 1. Introduction

There have been around seven pandemics in the last 130 years recorded in the health history of the world. They were in 2020 (China, COVID-19, first wave), 2009 (Mexico, swine flu-H1N1), 2003 (China, SARS), 1968 (Hong Kong, H3N2 virus), 1957 (Chinese- Asian flu, H2N2), 1918 (the USA, Army camps with three extended waves) and 1889 (Turkestan-H2N2 virus with three prolonged waves). However, the Coronavirus has badly affected the world’s economy and public life [1]. The same applies to the Omani context, which was negatively affected by the Coronavirus pandemic, especially in relation to health and the economy [1,2]. In Oman, COVID-19 first surfaced on 24 February 2020, and now its probability creates psychological, social, economic and financial effects on the public in Oman [3]. To minimize the health hazards of COVID-19, the government of Oman has gradually reduced public activities. This brought about suspension in socio-economic activities, creating higher financial and psychological stress on the citizens, including students and faculty living under the fear of a human-sourced pandemic.

The primary industry affected by Coronavirus’s appearance in Oman is the educational sector [4,5]. This has forced the Omani government to close schools to avoid spreading it among students and teachers. However, unfortunately, the number of studies exploring employee resilience is minimal [5]. In addition, to the best of our knowledge, hardly any studies focus on employee resilience concerning employee performance in the Omani education sector, which has been hit hard by Covid-19. All these reasons have encouraged us to conduct the current study to address the recent call for further research to contribute to contextual [4,6,7] and knowledge gaps [8].

This study has contributed in multiple ways. First, this study is considered one of the first to explore the mediating role of employee resilience during the Coronavirus pandemic using the gulf context. Second, some factors mentioned in this study, particularly resilience, are rarely mentioned in educational studies or contexts. Thus, academic readers of this study may become curious and subsequently want to conduct similar studies of this kind to understand better the effects on employee performance in their contexts. Finally, this study has extended the research on COVID-19 beyond its medical field to investigate its impact on the educational industry. This indicates that the current study is unique in its focus and topic, which deals with the current situation to better understand its influence on schools.

The study is conducted to answer a fundamental research question: How can employee resilience and digitalization interact in a way that could help the firm improve employee performance for a challenging task during a turbulent time? Generally speaking, this research aims to study the indirect role of employee resilience in government schools in the Al-Suwaiq region of Oman, which was affected the most due to Covid-19. Furthermore, identifying mediating relationships would help educational supervisors to determine what motivates teachers and how to sustain their motivation and satisfaction during the Coronavirus pandemic in Oman. More importantly, the outcomes of this study are helpful for the decision-makers and officials at the Ministry of Education and the Directorate General of Education in Batinah North to ensure teacher performance during this time.

## 2. Literature Review and Theoretical Underpinning

### 2.1. The Broaden-and-Build Theory

The broaden-and-build theory suggests that adverse events such as the Covid-19 pandemic positively affect resilient employees to enhance their overall satisfaction, including job satisfaction and performance [4,6,9]. For instance, a study identified that employee resilience strengthens coping and well-being in the workplace [4,10]. Thus, the subsequent conceptual model of this research study presented in the hypothesis development section is grounded in the broaden-and-build theory, which integrates psychology, education and sociology domains.

### 2.2. Hypothesis Development

According to research study [8], task challenge with due organizational support is essential for schoolteachers to carry out and achieve work objectives that let them obtain academic success. During Covid-19, the task challenge is at a different level due to exceptional circumstances [11]. Therefore, the task challenge plays a key role during extraordinary circumstances, including the Covid-19 pandemic [2,11]. According to the literature, employee performance is the willingness to work integrated with the positive attitude of employees about their jobs [11,12]. We propose a direct and positive linkage between task challenge and employee resilience during Covid-19, consistent with the broaden-and-build theory [8,11]. The disappointments that the workforce may experience during the lockdown could make them more resilient employees. For instance, if teachers believe that more technical skills could help their school, they should be more motivated to deliver more modern online lectures, thus challenging themselves. This behaviour demonstrates better reliance at the workplace to secure limited jobs during the pandemic. So, we can postulate the following hypothesis:

**H1.** *Task challenge positively associated with employee resilience during the COVID-19 pandemic*.

Such diminished employee resilience among schoolteachers motivates them to learn new online teaching methods, thus steering teachers towards better performance [11]. In particular, employee resilience could be down to their psychological, adaptive and planned organizational determination to express this solid psychological connection [4,11,12] with their school teaching. Moreover, consistent with the broaden-and-build theory [8,13], teachers marked by high resilience may demonstrate better work performance. For example, teachers may join online classes from home one time, stay longer on MS Teams to respond to students, and put more effort into their teaching and learning activities. Considering these arguments, we hypothesize the following:

**H2.** *Employee resilience is positively associated with employee performance*.

Employee resilience is the ability to face or solve challenges that employees encounter in their daily lives, specifically in their workplace [2,4,14]. In this study, we also propose the mediating role of employee resilience. Task motivation leads to enhanced employee resilience, which generates teacher satisfaction in relation to their job. For example, suppose teachers get challenged to adopt new technology and deliver an online lecture with interactive videos. In that case, positive student feedback will cause them to perform their job better. Prior studies also predicted the mediating role of employee resilience [2,4]. Thus, we add to previous research by postulating the following mediating role of employee resilience:

**H3.** *Employee resilience mediates the relationship between task challenge and employee performance*.

Lately, research has suggested that digitalization enhances employee performance [11,12]. Furthermore, digitalization helps faculty members deal with school-related issues firmly during the pandemic [2,12,13]. Therefore, we propose the moderating effect of digitalization on the relationship between task challenge and employee resilience during the pandemic [1,11,14]. If a higher degree of organizational resilience is needed, digitalization can help teachers deal firmly with COVID-19-related hardships [2,4,11]. Therefore, we have postulated the following moderating role of digitalization–organizational support during the COVID-19 pandemic:

**H4.** *The positive relationship between task challenge and employee resilience is moderated by digitalization, such that the relationship is strengthened with a high degree of digitalization*.

Statistically speaking, the above hypotheses (H1 to H4) and theoretical arguments suggest the presence of a moderated mediation [15]. Digitalization is the critical factor in the indirect relationship between task motivation and diminished employee resilience [16,17]. Teachers who get support from the organization regarding the digital environment at the workspace are more satisfied and willing to perform. Thus, employee resilience has a powerful effect on connecting task motivation and work performance. Therefore, we have postulated the following moderated mediation mechanism:

**H5.** *The indirect relationship between task challenge and employee performance through diminished employee resilience is moderated by digitalization, such that this indirect relationship is strengthened when a teacher gets organizational support in terms of digitalization and vice versa*.

The conceptual framework presents the five hypotheses (see Figure 1).

## 3. Methodology

All teachers in government schools in the Al-Suwaiq region of Oman’s North Al Batinah province represent the current study population. We created an online questionnaire using Google forms. The questionnaire consists of two sections. The first section was used to collect demographics, and the second was used to collect the responses of selected variables. According to the official statistics, the total number of teachers is 2541 in the academic year 2019/2020 for regional schools. Thus, we applied a simple random technique for the known population [3,15]. Before pilot testing, the Arabic-translated questionnaire was sent to language and area experts to check content validity. Feedback was provided, and some changes were made accordingly, especially concerning the layout and time required to complete the questionnaire. Then the questionnaire was tested for reliability, and the Cronbach alpha was calculated. Response rates and the validity and reliability of the questionnaire are discussed in the following parts.

We collected data using the online Google forms survey option and emails. So, we emailed 200 respondents and sent messages to 1700 participants via WhatsApp. We didn’t use printed questionnaires to collect data because of the Coronavirus pandemic. We got a 23% response rate via WhatsApp. The response rate was 25% when we requested to complete a survey via email.

### 3.1. Research Instrument

A questionnaire was designed to answer the question of the study and test the hypotheses. In the current study, the questionnaire consists of two following parts. The first part contains general demographic information. This part is introductory and focuses on getting basic information about the participants, such as their years of experience, specialization, nationality, gender, salary and qualification. The second part consisted of closed items from the Likert scale.

The study used the measurement model fit indices, namely the CMIN/df, comparative fit index (CFI), root-mean-square error of approximation (RMSEA), and the Tucker–Lewis index (TLI) as prescribed by scholars [18,19] to report confirmatory factor analyses (CFA). After three tries, we achieved a better fit of the measurement model with a series of CFAs by linking the error terms of the similar constructs of observed variables (X2 = 1365.21; *p* ≤ 0.001; CMIN/df = 1.59; RMSEA = 0.04; CFI = 0.94 and TLI = 0.93). Accordingly, the TLI and CFI were above the threshold value of 0.90, and the RMSEA score was below the threshold value (i.e., 0.05). Furthermore, Harman’s single factor exhibited 27.64% shared variance, presenting that the CMV (common method variance) is statistically insignificant [20].

*Digitalization:* Four-item scale for student support was adapted from the study of [19]. The sample items are: “Without an information system. It would be challenging to do my job.”; and “I depend heavily on the information system to take care of my responsibilities”.

*Task Challenge:* Five-item scale for the task challenge was adapted from the study Fernet et al. [10]. The sample items are: “Because it is important for me to carry out this task.”; and “Because this task allows me to attain work objectives that I consider important”.

*Employee Resilience:* Six-item scale for employee resilience was adapted from the study [4]. The sample items are: “*I learn from mistakes at work and improve the way I do my job.*”; and “*I seek assistance to work when I need specific resources*”. For the scale of employee resilience, there were six items.

*Employee Performance:* The scale for teacher performance as a proxy of employee performance was adapted from the two studies [21,22]. The sample items are: “I try to be at work as often as I can”; and “My work is always of high quality”.

### 3.2. Outer Model Assessment

Table 1 describes the standardized factor loadings of items, average variance extracted (AVE), Cronbach’s alpha and composite reliability (CR). According to statisticians [18,23], all the loadings were above the threshold of 0.70. All the AVE values for digitalization, task challenge, employee performance and employee resilience were 0.711, 0.597, 0.860 and 0.565, respectively, and above the threshold level of 0.50. Same as the AVE, the CR values of all the constructs exceeded the threshold level of 0.70. The CR values for digitalization, task challenge, employee performance and employee resilience were 0.908, 0.881, 0.970 and 0.836, respectively. Moreover, all the values of Cronbach’s alpha exceed the threshold level of 0.70 [20].

### 3.3. Discriminant Validity

According to statisticians [23,24,25] criteria and heterotrait-monotrait (HTMT) analysis, the discriminant validity has been confirmed through the model comparison method. Table 2 demonstrates the discriminant validity in the model accordingly.

In Table 3, we also presented the HTMT analysis. Thus, our measurement model confirms the discriminant validity in the HTMT analysis as a value less than the threshold of 0.9 [25].

## 4. Data Analysis and Findings

### Correlation and Descriptives Analysis

The descriptive statistics and correlation coefficients are reported in Table 4.

We also conducted the descriptive analysis and found that most participants are Omani teachers, with 400 respondents (90.7%) and 41 non-Omani teachers (9.3%). In addition, the sample is a mixture of male (59.9%) and female (40.1%) participants. These teachers were teaching students at various levels, i.e., 20.6% teach grades from one to four, 42.6% teach grades five to nine, and 36.1% teach grades nine to ten. We also analyzed the teachers’ teaching experience and found that 5.2% have one to five years of experience. A total of 15.6% have six to ten years of experience, and most teachers in this study have over ten years of experience (79.1%). We asked for their salaries in local currency. We found that a teacher was getting a salary of fewer than five hundred rials (3.6%), 5.6% were getting compensation from five hundred to nine hundred Omani rials, and most government teachers get above nine hundred OMR (87.3%). Teachers were teaching various subjects. Around 13.4% taught English, 23.1% Arabic, 8.8% Islamic studies, 3.4% History, 4.5% Geography, 11.8% Science, 7.5% Life skills, 9.5% Mathematics, 1.4% Sport, 3.2% Music and 13.4% other subjects; all data are from teachers who responded to our survey.

Table 5 presents the mediation results [15,26] obtained from the process macro (Model 4). After controlling for a few variables (nationality, gender, age, qualification, teaching grade, experience and teaching subject), we found that the task challenge–employee performance nexus was statistically significant and positive (β = 0.214, *p* < 0.001), which substantiated our first hypothesis. Thus, these findings concerning the relationship between task performance and employee performance are consistent with preceding research findings [21,22]. The results also provided the foundation to test our moderated-mediation mechanism [27]. We also found a statistically significant impact of task-challenge during the pandemic on employee resilience (β = 0.429, *p* < 0.001), which substantiated our second hypothesis. The findings of the relationship between task challenge and employee resilience are consistent with previous research studies [3,4]. Finally, the mediation results reveal that the employee residence mediates the relationship between task challenge and employee performance. The mediation test for the data set shows an effect size of 0.120 for the indirect relationship between task challenge and employee performance measured through employee resilience. The (confidence interval) CI did not include zero value [M1: 0.039, 0.206] and confirmed the presence of mediation, which substantiated our second hypothesis.

The moderated-mediation results in Table 6 tell us a unique story about digitalization. We assumed that digitalization could play a vital role during the pandemic as this was the direct organizational support for the teachers during the unusual circumstances caused by the Covid-19 outbreak. We controlled variables (nationality, gender, age, qualification, teaching grade, experience and teaching subject) in the light of previous research [2,4,19,22]. The positive impact of digitalization on employee performance is well established [16,17,19]. Consistently, our study found the direct effect of digitalization on employee performance (β = 0.223, *p* < 0.001) was statically significant (See Table 5). These findings motivated us to focus on our results in Table 6 to reveal a moderated-mediation mechanism of digitalization through employee resilience results in employee performance.

The first part of the output presented in Table 6 shows the direct effects of digitalization, task challenge and interaction (Task Challenge x Digitalisation). We observed that the interaction term’s beta was statistically significant (β = 0.089, *p* < 0.05). Thus, the results suggest that digitalization moderates the effect of task challenges on employee resilience. Therefore, higher organizational support in digitalization should generate more employee resilience. However, the moderated mediation was not confirmed as the zero falls between the upper and lower bounds of the confidence interval. Therefore, Hypothesis 6 was not substantiated.

## 5. Discussion

The study was to answer the research question: how can employee resilience and digitalization interact to help the firm improve employee performance for a challenging task during a turbulent time? There are three key reasons for answering this. First, the challenging task during turbulence needs a firm to focus on generating employee resilience. Second, digitalization could help to retain customers and enable employees to work from home. Finally, if the firm can survive during a crisis, then the firm has a higher chance of growth once that turbulent time is over.

First, the research found that task-challenge influence employee-performance. These findings are consistent with preceding research findings [10,21,22]. The reason is that in a challenging task, competent employees learn and improve their performance during times of crisis. Similarly, teachers took that challenge and helped the university to deliver online education. Second, we also found that a statistically significant task challenge generates employee resilience. Accordingly, teachers worked to enhance their resilience by taking online classes using various software such as MS Teams and Google Meet. They also recorded lectures and took exams online using the latest technology. These findings were consistent with previous research studies [4,7,19]. Thirdly, the mediation results reveal that employee resilience mediates the relationship between task challenge and employee performance. Fourthly, the moderated-mediation results tell us a unique story about digitalization’s role during periods of turbulence. For instance, it was quite impossible to continue university classes conventionally during the pandemic. The digital university portals, online attendance and mobile applications for student records played a pivotal role in achieving the goals of employees and universities. The positive impact of digitalization on employee performance is well established [19], but in our study, we explained its moderating role. Nevertheless, the moderated mediation of digitalization and employee resilience was unconfirmed, and the result was inconsistent with preceding studies [4,10,19].

### Limitations and Future Research

There were limitations in terms of time, place, participants and analysis skills. In terms of time, the current study collected data only during the Coronavirus pandemic, so future time-lag-based studies could be useful. In terms of location, the study was limited to government schools. Therefore, the outcomes cannot be generalized to private schools and higher education institutes. Finally, in terms of participants, the study was limited to teachers. Other school community members, such as principals, students and support staff, were omitted. Their views and opinions would enrich the discussion of the current study if they were involved [2,5,6,28], but this was not possible due to the limited time and resources available to conduct this type of scientific research [23,24,28].

The limitations discussed in the preceding section may open the door for future research studies on the same topic. For instance, we found gender and qualification were impacting the digitalization process. So, a group-based analysis could help to see the moderating role of gender and education level during the digitalization process of the firm [7,16,19]. This study has applied a questionnaire to collect data. Future research studies may use other strategies, such as case studies or action research, to broaden and deepen the factors affecting employee performance [21,22,28].

## 6. Conclusions and Implications

The outcomes of the current study can assist decision-makers and senior officials in better understanding the effects of the factors included in this study on employee performance, digitalization and resilience during unusual circumstances. Maintaining teachers’ performance during this Coronavirus pandemic is not easy. Thus, as a suggestion, the mediated moderation mechanism mentioned in this study can make the necessary plans and precautions to protect students and school staff. Then, it will be possible to take the required procedures and actions to keep teachers highly motivated and satisfied during the turbulent environment.

## Figures and Tables

**Figure 1 behavsci-13-00119-f001:**
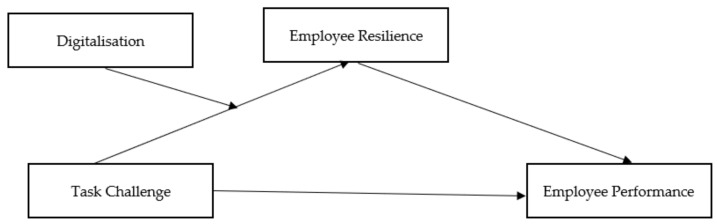
A conceptual framework for employee resilience.

**Table 1 behavsci-13-00119-t001:** Instrument, standardized loadings, AVE, CR and Cronbach’s alpha.

Sr. No.	Factor	Items	Estimate	CR	AVE	Cronbach’s alpha
1	Digitalisation	DIG1	0.815	0.908	0.711	0.864
		DIG2	0.868			
		DIG3	0.892			
		DIG4	0.794			
2	Task challenge	TCE1	0.703	0.881	0.597	0.785
		TCE2	0.815			
		TCE3	0.815			
		TCE4	0.809			
		TCE5	0.713			
3	Employee performance	EPE1	0.950	0.970	0.860	0.940
		EPE2	0.960			
		EPE3	0.970			
		EPE4	0.860			
		EPE5	0.890			
4	Employee resilience	ERE1	0.570	0.836	0.565	0.735
		ERE2	0.570			
		ERE3	0.630			
		ERE4	0.650			
		ERE5	0.860			
		ERE6	0.720			

**Table 2 behavsci-13-00119-t002:** Discriminant validity.

Variables	MEAN	SD	1	2	3	4
1. Digitalization	3.700	0.800	0.760			
2. Task challenge	3.620	0.910	0.140	0.790		
3. Employee performance	3.200	1.240	0.150	0.480	0.930	
4. Employee resilience	3.450	0.980	0.180	0.610	0.390	0.820

**Table 3 behavsci-13-00119-t003:** HTMT analysis.

Variables	1	2	3	4
1. Digitalization				
2. Task challenge	0.190			
3. Employee performance	0.216	0.564		
4. Employee resilience	0.182	0.618	0.388	

**Table 4 behavsci-13-00119-t004:** Correlations.

Variable Name	Mean	SD	1	2	3	4
1. Digitalization	2.36	0.61	1			
2. Employee resilience	1.84	0.46	0.328 **	1		
3. Task challenge	1.89	0.57	0.303 **	0.548 **	1	
4. Employee performance	2.32	0.70	0.323 **	0.364 **	0.356 **	1

** *p* < 0.01; N = 441; NS = not significant; SD: standard deviation. Alpha is presented in parenthesis.

**Table 5 behavsci-13-00119-t005:** Mediation analysis (Process Macro Model 4).

Variables	Employee Resilience (M1)	Employee Performance (M2)
Task challenge	0.429 **	0.214 **
ER	-	0.280 *
Digitalisation	-	0.223 **
Nationality	−0.120	0.005
Gender	−0.068	−0.008
Age	−0.007	0.039
Qualification	−0.008	0.039
Teaching grade	0.012	0.039
Experience	−0.016	−0.008
Teaching subject	−0.006	−0.045
R^2^	0.311 *	0.238 *
Indirect effect (Task challenge–employee resilience–employee performance)		
Effect size		0.120
Bootstrap SE		0.042
LLCI		0.039
ULCI		0.206

Note(s): EP: employee performance; ER: employee resilience, ** *p* < 0.01; * *p* < 0.05. *n* = 441. LLCI: lower limit confidence interval; ULCI: upper limit confidence interval.

**Table 6 behavsci-13-00119-t006:** Moderated-mediation analysis (Process Macro Model 7).

Variables	Employee Resilience (M1)		Employee Performance (M2)	
Task challenge	0.0174		0.253 **	
ER	-		0.344 **	
Digitalisation	0.037		0.223 **	
TC x digitalisation	0.089 *		-	
Nationality	−0.149		0.099	
Gender	−0.057		−0.235 **	
Age	−0.012		−0.037	
Qualification	−0.015		0.223 **	
Teaching grade	−0.004		−0.061	
Experience	−0.004		−0.073	
Teaching subject	−0.004		0.187	
*R* ^2^	0.343 **		0.207 **	
	Effect size	Bootstrap SE	LLCI	ULCI
Conditional indirect effect of perceived task challenge on employee resilience				
−1 SD		0.035	0.050	0.191
Mean		0.036	0.058	0.204
+SD		0.047	0.067	0.251
Index of moderated mediation				
Digitalization		0.0240	−0.0127	0.0834

Note(s): EP: employee performance; ** *p* < 0.01. * *p* < 0.05. SD: standard deviation; LLCI: lower limit confidence interval; ULCI: upper limit confidence interval. *n* = 441.

## Data Availability

The data are not publicly available due to privacy or ethical restrictions.

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
