# Peer review of "Task Challenge and Employee Performance: A Moderated Mediation Model of Resilience and Digitalization"

_behavsci, 2023, doi:10.3390/bs13020119_

Round 1

Reviewer 1 Report

I realize that great work and time have been devoted to this paper. It has a lot of strengths, but I think that some changes should be recommended. 

Title: the title does not adequately reflect the content of the paper. Please, try to change it to better inform the readers about the relationships between the variables you test and also inform them about the quality of your sample.

Abstract:

Less information appears in the abstract. Maybe expanded by adding the most relevant findings. Please, consider that the abstract is the unique part of your paper that most of the readers could read. Hence, more information would be better.  

Keywords: it is better to enlist your keywords alphabetically. Do not use keywords already captured in the title of the manuscript.

Introduction

The literature revision has some references that are too old. Besides citing some papers from 2001, you can consider some relevant papers on the topic RECENTLY published in other Journals. There are some Journals that suggest a high percentage of references published during the last five years. The introduction is too brief. The most relevant findings related to the constructs under study should be summarized.

Methodology

The Instruments or Questionnaires section needs more information. Please, some examples of items should be provided to the readers. If you can, please inform me about previous studies where the same instrument has been used and the reliability obtained in that research.

Results

Throughout the paper, but mostly in the results section, there are many abbreviations. This renders it too difficult for readers to understand the content. Related to your results, I would suggest you clarify for readers your results providing a figure with the coefficients.

Discussion:

First of all, try to adjust your conclusions to the findings better. Or to say in other words, please try to justify more clearly the connection between your conclusions and your findings.

Finally, a section related to limitations, future lines of investigation, and the principal contributions of the research could be attractive. Your paper has a lot of relevant implications for educators, psychologists, society, and policymakers, but you need to elaborate more on this topic.

Author Response

Comment: I realize that great work and time have been devoted to this paper. It has a lot of strengths, but I think that some changes should be recommended. 

Title: the title does not adequately reflect the content of the paper. Please, try to change it to better inform the readers about the relationships between the variables you test and also inform them about the quality of your sample.

Reply: Title has been updated. 

Abstract:

Less information appears in the abstract. Maybe expanded by adding the most relevant findings. Please, consider that the abstract is the unique part of your paper that most of the readers could read. Hence, more information would be better.  

Reply: Abstract has been updated. 

Keywords: it is better to enlist your keywords alphabetically. Do not use keywords already captured in the title of the manuscript.

Introduction

The literature revision has some references that are too old. Besides citing some papers from 2001, you can consider some relevant papers on the topic RECENTLY published in other Journals. There are some Journals that suggest a high percentage of references published during the last five years. The introduction is too brief. The most relevant findings related to the constructs under study should be summarized.

Reply: This section has been updated as per the comments. 

Methodology

The Instruments or Questionnaires section needs more information. Please, some examples of items should be provided to the readers. If you can, please inform me about previous studies where the same instrument has been used and the reliability obtained in that research.

Results

Throughout the paper, but mostly in the results section, there are many abbreviations. This renders it too difficult for readers to understand the content. Related to your results, I would suggest you clarify for readers your results providing a figure with the coefficients.

Reply: This section has been updated as per the comments. 

Discussion:

First of all, try to adjust your conclusions to the findings better. Or to say in other words, please try to justify more clearly the connection between your conclusions and your findings.

Reply: This section has been updated as per the comments. 

Finally, a section related to limitations, future lines of investigation, and the principal contributions of the research could be attractive. Your paper has a lot of relevant implications for educators, psychologists, society, and policymakers, but you need to elaborate more on this topic.

Reply: This section has been updated as per the comments. 

Reviewer 2 Report

The effort put in is to be commended. However, recent literature should be used to fill the research gap. The research problem should be clarified further. The methodology is not entirely clear. The study population, sample selection, and sample size should all be clearly stated. There were no analytical methods described. Further language editing and formatting are required. Some comments are given on the paper, pl refer to them.

Author Response

Reply:  The changes have been incorporated in light of the comments. 

Reviewer 3 Report

I reviewed your work titled “A Moderated Mediation Model of Employee Resilience during COVID 19 Pandemic”. You have examined the process in education during the covid-19 pandemic period. You have written well the contributions of the study in the Introduction section. Congratulations. This study is important, but I would like to point out that there are some important corrections. I have tried to list them one by one below.

- Rearrange the title of the study by adding other scales you used.

- Enter the beginning of the abstract with one or two sentences about the subject and problem of the study.

- In the summary, write in one sentence how and with what analysis you did the study online, cross-sectional, with how many participants.

- A student or students? A student deserves a thank you at the end of the study.

- In the last paragraph of the introduction of the study, write a research question about why you did this study and evaluate this question in the discussion section.

- The resources you use in the literature section are not enough. I recommend that you access more resources.

- Since this study is cross-sectional, it would be better if you use the concepts of relationship or association instead of impact or effect.

- Theoretical framework should be drawn better. So the total relationship should also be shown. It will be better understood if you draw it as Hayes used in model 7.

- It is not clear how the scales are scored. For example, you could write: As the score obtained from the scale increases, employee resilience increases.

- How did you analyze the data of the study? Did you do factor analysis of EFA or CFA? Which of these you have done, you need to put the results in the article.

- The correlation table was misplaced or shifted there.

- I recommend you to use two-way interaction slope to show the moderation result. You can find it at this link. http://www.jeremydawson.co.uk/slopes.htm

- Discuss in the article under the heading Discussion, in line with the sub-headings of direct, indirect, moderation and evaluation of the research question, in line with the results of the current literature and hypotheses.

- Your Limitations header must be separate

- Conclusions and Implications should be a separate title at the end.

Author Response

I reviewed your work titled “A Moderated Mediation Model of Employee Resilience during COVID 19 Pandemic”. You have examined the process in education during the covid-19 pandemic period. You have written well the contributions of the study in the Introduction section. Congratulations. This study is important, but I would like to point out that there are some important corrections. I have tried to list them one by one below.

- Rearrange the title of the study by adding other scales you used.

Reply:  The changes have been incorporated. 

- Enter the beginning of the abstract with one or two sentences about the subject and problem of the study.

- In the summary, write in one sentence how and with what analysis you did the study online, cross-sectional, with how many participants.

- A student or students? A student deserves a thank you at the end of the study.

Reply:  The changes have been incorporated. 

  • In the last paragraph of the introduction of the study, write a research question about why you did this study and evaluate this question in the discussion section.

- The resources you use in the literature section are not enough. I recommend that you access more resources.

- Since this study is cross-sectional, it would be better if you use the concepts of relationship or association instead of impact or effect.

Reply:  The changes have been incorporated. 

- Theoretical framework should be drawn better. So the total relationship should also be shown. It will be better understood if you draw it as Hayes used in model 7.

- It is not clear how the scales are scored. For example, you could write: As the score obtained from the scale increases, employee resilience increases.

- How did you analyze the data of the study? Did you do factor analysis of EFA or CFA? Which of these you have done, you need to put the results in the article.

- The correlation table was misplaced or shifted there.

- I recommend you to use two-way interaction slope to show the moderation result. You can find it at this link. http://www.jeremydawson.co.uk/slopes.htm

- Discuss in the article under the heading Discussion, in line with the sub-headings of direct, indirect, moderation and evaluation of the research question, in line with the results of the current literature and hypotheses.

- Your Limitations header must be separate

Reply:  The changes have been incorporated. 

- Conclusions and Implications should be a separate title at the end.

Reply:  The changes have been incorporated. 

Round 2

Reviewer 3 Report

You said all of them were incorporated, but the following were not done. Otherwise, it is not possible for me to approve this work.

- Enter the beginning of the abstract with one or two sentences about the subject and problem of the study.

- In the summary, write in one sentence how and with what analysis you did the study online, cross-sectional, with how many participants.

- A student or students? A student deserves a thank you at the end of the study.

- In the last paragraph of the introduction of the study, write a research question about why you did this study and evaluate this question in the discussion section.

- The resources you use in the literature section are not enough. I recommend that you access more resources.

- Since this study is cross-sectional, it would be better if you use the concepts of relationship or association instead of impact or effect.

- Theoretical framework should be drawn better. So the total relationship should also be shown. It will be better understood if you draw it as Hayes used in model 7.

- It is not clear how the scales are scored. For example, you could write: As the score obtained from the scale increases, employee resilience increases.

- How did you analyze the data of the study? Did you do factor analysis of EFA or CFA? Which of these you have done, you need to put the results in the article.

- The correlation table was misplaced or shifted there.

- I recommend you to use two-way interaction slope to show the moderation result. You can find it at this link. http://www.jeremydawson.co.uk/slopes.htm

- Discuss in the article under the heading Discussion, in line with the sub-headings of direct, indirect, moderation and evaluation of the research question, in line with the results of the current literature and hypotheses.

- Your Limitations header must be separate

- Conclusions and Implications should be a separate title at the end.

Author Response

Dear Editor, Find attached the updated manuscript. Following is the reply, and changes are made accordingly. 

- Enter the beginning of the abstract with one or two sentences about the subject and problem of the study.

Reply: Sentence added about employee's resistance. 

- In the summary, write in one sentence how and with what analysis you did the study online, cross-sectional, with how many participants.

Reply: Sentence added in the abstract section.  

- A student or students? A student deserves a thank you at the end of the study.

Reply: Sentence deleted. however, in the acknowledgement section, we can add a note of thanks.

- In the last paragraph of the introduction of the study, write a research question about why you did this study and evaluate this question in the discussion section.

Reply:  In the last paragraph of the introduction of the study, a research question is added, and in the discussion section, we have added why we did this study and evaluated this question in the discussion section.

- The resources you use in the literature section are not enough. I recommend that you access more resources.

Reply: Latest literature/references have been added.

- Since this study is cross-sectional, it would be better if you use the concepts of relationship or association instead of impact or effect.

Reply: H1 and H2 have been updated.

- Theoretical framework should be drawn better. So the total relationship should also be shown. It will be better understood if you draw it as Hayes used in model 7.

Reply: Framework is updated.

- It is not clear how the scales are scored. For example, you could write: As the score obtained from the scale increases, employee resilience increases.

Reply: Not applicable

- How did you analyze the data of the study? Did you do factor analysis of EFA or CFA? Which of these you have done, you need to put the results in the article.

Reply: Analysis has been added. See tables 2,3,4

- The correlation table was misplaced or shifted there.

Reply:  table has been moved.

- I recommend you to use two-way interaction slope to show the moderation result. You can find it at this link. http://www.jeremydawson.co.uk/slopes.htm

Reply thanks for the recommendation, but the interaction effect has been well explained using data. so we are not making this change. 

- Discuss in the article under the heading Discussion, in line with the sub-headings of direct, indirect, moderation and evaluation of the research question, in line with the results of the current literature and hypotheses.

Reply: Discussion has been enhanced.  

- Your Limitations header must be separate

Reply: separated. 

- Conclusions and Implications should be a separate title at the end.

Reply: header added

Round 3

Reviewer 3 Report

I couldn't see confirmatory factor analysis fit values such as RMSEA, GFI, AGFI, RMR, and SRMR of the measurement model. Why don't you at least write those values in the text?

Author Response

I couldn't see confirmatory factor analysis fit values such as RMSEA, GFI, AGFI, RMR, and SRMR of the measurement model. Why don't you at least write those values in the text?

Reply:  CFA analysis has been added.